# The Optimal Sample Complexity of Linear Contracts

**Mikael Møller Høgsgaard** [1] [2]

## Abstract

In this paper, we settle the problem of learning optimal linear contracts from data in the offline setting, where agent types are drawn from an unknown distribution and the principal's goal is to design a contract that maximizes her expected utility. Specifically, our analysis shows that the simple Empirical Utility Maximization (EUM) algorithm yields an $\varepsilon$-approximation of the optimal linear contract with probability at least $1 - \delta$, using just $O(\ln(1/\delta)/\varepsilon^2)$ samples. This result improves upon previously known bounds and matches a lower bound from (Dütting et al., 2025b) up to constant factors, thereby proving its optimality. Furthermore, our result establishes the stronger guarantee of uniform convergence: the empirical utility of every linear contract is an $\varepsilon$-approximation of its true expectation with probability at least $1 - \delta$, using the same optimal $O(\ln(1/\delta)/\varepsilon^2)$ sample complexity.

## 1. Introduction

A central problem in algorithmic contract theory is to design incentives for agents whose characteristics are unknown and must be learned from data. Consider a digital music platform looking to introduce a new royalty model (contract). Each independent musician (agent) on the platform has a private type, reflecting their creative process and cost of effort, drawn from a population-level distribution that is unknown to the platform. Before implementing a site-wide change of royalty model, the platform runs a pilot program with a small sample of musicians. In this program, it tests several new revenue-sharing contracts and gathers detailed data on their resulting song downloads and streaming engagement. Based on this sample, the platform aims to learn an improved royalty model that optimizes its profits by motivating its entire community of artists.

This "pilot study" is an example of the scenario formalized in the recent seminal work of (Dütting et al., 2025b), which establishes a sample-based learning framework for designing an optimal contract from a finite dataset of fully-profiled agents. This framework complements other established models in the literature, each suited for different scenarios. For instance, the Bayesian setting models situations where the principal has full distributional knowledge, ideal for full-information and static scenarios. In contrast, online learning models address dynamic settings where a contract must be adapted through repeated, real-time interactions with agents. The framework of (Dütting et al., 2025b) thus captures yet another important real world scenario, the finite sample setting.

More formally, (Dütting et al., 2025b) consider the following framework, which we adopt (almost) and now formally define. The environment is fixed by a set of $n$ actions an agent can take, indexed by $[n] = \{1, \ldots, n\}$, and $m \geq 2$ possible outcomes, indexed by $[m] = \{1, \ldots, m\}$. For each outcome $j \in [m]$, the principal receives a known (both to the principal and the agent), fixed reward $r_j \geq 0$. It is assumed that $r_1 = 0$ and there is at least one outcome with a positive reward. An agent is characterized by a private type $\theta = (f, c)$ (i.e., unknown to the principal during live interaction), which consists of two components:

- A production function $f = (f_1, \ldots, f_n)$, where each $f_i$ is a probability distribution over the $m$ outcomes. Specifically, $f_{i,j}$ is the probability of observing outcome $j$ if the agent chooses action $i$.
- A cost vector $c = (c_1, \ldots, c_n)$, where $c_i \geq 0$ is the personal cost for the agent to take action $i$. We assume that action 1 is an outside option with zero cost, i.e., $c_1 = 0$.

The principal designs a contract, which is a payment vector $t = (t_1, \ldots, t_m)$ where $t_j \geq 0$. If outcome $j$ occurs, the agent is paid $t_j$. Given a contract $t$, an agent of type $\theta$ will choose an action $i \in [n]$ to maximize their own expected utility:

$$u_a(\theta, t, i) = \sum_{j=1}^{m} f_{i,j} t_j - c_i \qquad (1)$$

The principal's utility depends on which action the agent takes. Assuming the agent breaks ties in the principal's

---

[1]Department of Statistics, University of Oxford, United Kingdom [2]Department of Computer Science, Aarhus University, Denmark. Correspondence to: Mikael Møller Høgsgaard <hogsgaard@cs.au.dk>.

*Proceedings of the 43rd International Conference on Machine Learning*, Seoul, South Korea. PMLR 306, 2026. Copyright 2026 by the author(s).

favor, the agent chooses the action $i^*(\theta, t)$ that maximizes the principal's utility from the set of the agent's own best actions (those maximizing Equation (1)).[1] The principal's utility for a given type $\theta$ is then:

$$u_p(\theta, t) = \sum_{j=1}^m f_{i^*(\theta,t),j}(r_j - t_j)$$

Finally, we define the learning objective. The principal's goal is to find a contract $t$ that maximizes the expected utility $u_p(\mathcal{D}, t) = \mathbb{E}_{\theta \sim \mathcal{D}}[u_p(\theta, t)]$ over an unknown distribution of agent types $\mathcal{D}$. The learning model of (Dütting et al., 2025b) assumes the principal has access to a dataset $\mathbf{S} = \{\theta_1, \ldots, \theta_s\}$ of $s$ i.i.d. samples from $\mathcal{D}$, and for each sample $\theta_i \in \mathbf{S}$, the principal is given the full type (i.e., the production function $f^{(i)}$ and cost vector $c^{(i)}$), which allows the principal to simulate the agent's behavior and compute $u_p(\theta_i, t)$ for any candidate contract $t$.

As described, the basic framework of (Dütting et al., 2025b) assumes the principal receives samples of full agent types. We will, however, make a slightly weaker assumption, namely that the principal only has oracle access to compute the empirical utility $u_p(\mathbf{S}, t) = \frac{1}{s}\sum_{i=1}^s u_p(\theta_i, t)$ for any candidate contract $t$, but is not given the specific type of the sampled agent nor their set of actions. This assumption is weaker than the basic assumption made in (Dütting et al., 2025b) and still captures the offline setting, where the principal first gathers information to compute $u_p(\mathbf{S}, t)$ for any $t$ and then does not interact with the agents again. To the best of our knowledge, some of the results from (Dütting et al., 2025b) also hold in this weaker setting; we will comment on this when in order. Furthermore, following (Dütting et al., 2025b), we assume oracle access to $u_p(\mathbf{S}, t)$ in order to isolate the statistical question studied here: how many samples are needed for empirical utility to generalize to expected utility. This abstracts away from the computational problem of evaluating or optimizing empirical utilities, which is an important line of work in its own right (Babaioff et al., 2006; Dütting et al., 2021b;a; 2023a; Ezra et al., 2024; Dütting et al., 2024a; Deo-Campo Vuong et al., 2024; Dütting et al., 2025a; 2024b;c), but is not the focus of this paper.

Within this framework, (Dütting et al., 2025b) established a link between the sample complexity of learning a contract class and its pseudo-dimension, a combinatorial complexity measure. While their work provides general tools for analysis, the precise sample complexity remained unsolved for one of the most fundamental classes of contracts: linear contracts. These contracts formalize one of the most intuitive and simple ways to align incentives, namely the principal and agent share the realized reward according to a fixed percentage, as in music royalties, franchise royalties,

and sales commissions. Despite linear contracts' simplicity, which makes them appealing from a practical standpoint, they are also known to exhibit robustness to unknown agent actions (Carroll, 2015) and to be able, under certain conditions, to approximate the performance of fully optimal, yet more complex, contracts (Dütting et al., 2019).

In this paper, we precisely characterize the sample complexity for learning an $\varepsilon$-approximation of the optimal linear contract. Specifically, we show that the simple Empirical Utility Maximization (EUM) algorithm, choosing a contract within the linear contracts that maximizes the empirical utility, yields an optimal contract up to an additive $\varepsilon$-error with probability $1 - \delta$ given $O(\ln(1/\delta)/\varepsilon^2)$ samples, which is tight up to constant factors due to a lower bound of (Dütting et al., 2025b). Furthermore, we show the same optimal sample complexity bound for the harder problem of learning the class of linear contracts uniformly, that is, ensuring the empirical and expected utilities are simultaneously $\varepsilon$-close for all linear contracts.

Our tighter bound comes from a more direct analytical path leveraging key properties of linear contracts. While the general theory of (Dütting et al., 2025b) relies on the combinatorial abstraction of pseudo-dimension (See Theorem B.1), our proof uses a "first-principles" chaining argument. The key technical insight is to exploit the inherent monotonic structure of the expected reward of linear contracts. This property allows for the construction of a fine-grained net over the contract space, enabling a chaining argument that yields the optimal sample complexity. This approach handles the discontinuities and non-monotonicity of the utility function. In doing so, we demonstrate how exploiting the specific structure of a contract class can lead to optimal results where the general-purpose tools of previous work did not.

To describe the results, we define the class of linear contracts as $\mathcal{C}_{linear} = \{\alpha r \mid \alpha \in [0, 1]\}$ for a fixed $r \in [0, 1]^m$, where we write $\alpha$ as shorthand for a contract in $\mathcal{C}_{linear}$, and we will also interchange between $\mathcal{C}_{linear}$ and $[0, 1]$. Formally, we show the following theorem, which is the main result of this paper.

**Theorem 1.1** (Main Result). *Let $\mathcal{D}$ be an unknown distribution over agent types, $r \in [0, 1]^m$ be a reward vector, and let $\varepsilon > 0$ and $\delta \in (0, 1)$ be given. Then, for $s \geq 3456 \ln(4/\delta)/\varepsilon^2$, with probability at least $1 - \delta$ over $\mathbf{S} \sim \mathcal{D}^s$, it holds for any $\alpha \in \mathcal{C}_{linear}$:*

$$|u_p(\mathcal{D}, \alpha) - u_p(\mathbf{S}, \alpha)| \leq \varepsilon.$$

We emphasize that, despite the one-dimensional parametrization of linear contracts by $\alpha$, the utility function $u_p(\theta, \alpha)$ need not be continuous in $\alpha$: small changes in $\alpha$ may change the agent's optimal action discontinuously see e.g. Figure 1.

---

[1]This principal-favoring tie-breaking convention seems to be standard in the algorithmic contract theory literature; see, e.g., (Dütting et al., 2019; Xiao et al., 2020; Castiglioni et al., 2021; Dütting et al., 2024b; 2025b).

We note that the main theorem gives a uniform convergence bound for learning the difference between the empirical and expected utility for the class of linear contracts, with sample complexity independent of the number of actions $n$ and the number of outcomes $m$. Actually the sample complexity is assympotically equivalent to the bound for any single fixed contract. This is desirable, as we assume that the principal does not know the number of actions $n$, and the number of actions and outcomes $m$ could be large.[2]

Furthermore, the uniformity of the bound allows the principal not only to learn the utility of the optimal contract up to an additive $\varepsilon$ factor, but also to compare the utility of any two contracts and assess which is better, up to $\varepsilon$ precision.

It is also worth noting that the bound, up to constants, is the same as if one wanted to guarantee that the empirical utility of a single contract is $\varepsilon$-close to the expected utility of that contract. Thus, guaranteeing that the empirical utility of any (or one) contract is $\varepsilon$-close to its expected utility requires, up to constant factors, the same number of samples.

As a corollary of Theorem 1.1, it follows that the simple EUM algorithm achieves the optimal sample complexity.

**Corollary 1.2** (EUM Optimal Sample Complexity). *Let $\mathcal{D}$ be an unknown distribution over agent types, $r \in [0,1]^n$ be a reward, and let $\varepsilon > 0$ and $\delta \in (0,1)$ be given. Then, for $s \geq 6912 \ln{(4/\delta)}/\varepsilon^2$ with probability at least $1 - \delta$ over $\mathbf{S} \sim \mathcal{D}^s$, it holds that Algorithm 1[Empirical Utility Maximization algorithm] returns a contract $\hat{\alpha} \in \mathcal{C}_{linear}$ such that:*

$$u_p(\mathcal{D}, \hat{\alpha}) \geq \sup_{\alpha \in \mathcal{C}_{linear}} u_p(\mathcal{D}, \alpha) - \varepsilon.$$

*and $\hat{\alpha}$ is found by asking $O(1/\varepsilon)$ queries to the oracle for $u_p(\mathbf{S}, \cdot)$.*

To the best of our knowledge, Theorem 1.1 and Theorem 1.2 are the first results in the statistical setting introduced by (Dütting et al., 2025b) to show optimal sample complexity for a non-trivial class of contracts, both for uniform convergence and for learning the optimal contract up to an $\varepsilon$ error. Thus, we view these results as a step towards understanding the optimal sample complexity of learning other contracts in the setting of (Dütting et al., 2025b). We remark that we did not attempt to optimize the constants in the bound of Theorem 1.1 and Theorem 1.2.

### 1.1. Related Work

The study of contracts has a rich history in economics, with seminal contributions from (Holmström, 1979) and (Gross-

man & Hart, 1983). The importance of the field, as well as the foundational work of Oliver Hart and Bengt Holmström, was highlighted when the Nobel Prize in Economics in 2016 was awarded to Oliver Hart and Bengt Holmström for their work on contract theory (Royal Swedish Academy of Sciences, 2016).

Although contract design has its roots in economics, it has also garnered significant interest at the intersection of economics and computer science, particularly with the emergence of algorithmic contract design. The study of algorithmic contracts encompasses several distinct settings and aspects (some of which include, but are not limited to): The computational facets of contract design (Babaioff et al., 2006; Dütting et al., 2021b;a; 2023a; Ezra et al., 2024; Dütting et al., 2024a; Deo-Campo Vuong et al., 2024; Dütting et al., 2025a; 2024b;c); the Bayesian setting, where the distribution of agent types is known (Guruganesh et al., 2021; Alon et al., 2021; Castiglioni et al., 2022; Alon et al., 2023; Guruganesh et al., 2023; Castiglioni et al., 2025); and the online setting, where the principal interacts sequentially with agents, receiving only bandit feedback, and must design contracts on the fly (Ho et al., 2014; Cohen et al., 2022; Dütting et al., 2023b; Zhu et al., 2023; Bacchiocchi et al., 2024; Chen et al., 2024; Scheid et al., 2024; Burkett & Rosenthal, 2024). This paper focuses on the offline setting introduced by (Dütting et al., 2025b), and we refer the reader to their work for a more comprehensive comparison of this setting with other paradigms. Furthermore the statistical setting of (Dütting et al., 2025b) is related to a line of work in the intersection of game theory and learning theory, which studies the sample complexity in games and mechanism design, see e.g. (Cole & Roughgarden, 2014; Morgenstern & Roughgarden, 2015; Balcan et al., 2018; 2021; Hanneke et al., 2026). In this work we also uses tools from learning theory, such as uniform convergence, chaining arguments, empirical risk minimization/utility maximization, see e.g. (Shalev-Shwartz & Ben-David, 2014; Wainwright, 2019).

### 1.2. Comparison to Previous Work

In the offline setting, (Dütting et al., 2025b) shows two upper bounds on the sample complexity of learning the best linear contract: Theorem 4.1 (combined with Theorem 3.7) and Theorem 5.4. These theorems show that either $O((\ln{(1/\varepsilon)} + \ln{(1/\delta)})/\varepsilon^2)$ or $O((\ln{(n)} + \ln{(1/\delta)})/\varepsilon^2)$ samples are sufficient to learn the best linear contract up to an additive error of $\varepsilon$ with probability at least $1 - \delta$.

Some comments are in order regarding these two bounds. Both bounds are proven by upper-bounding the pseudo-dimension $d$ of a class of contracts and then applying Theorem 3.7 in (Dütting et al., 2025b), which, given such a bound, gives a sample complexity of $O((d + \ln{(1/\delta)})/\varepsilon^2)$.

In the first case, the bound on the pseudo-dimension is not

---

[2]We chose not to pursue the results for action spaces of infinite size to keep the presentation simple, as that case requires more technical scrutiny. For instance, the optimal action of the agent has to be defined differently, as the maximum may not be attained by an action. We, however, believe that the argument is not inherent to the finite action space case.

on the space of linear contracts itself but is instead on a discretization of the contract space consisting of multiples of $\varepsilon$, where this discretization preserves a good approximation of the best contract. This gives a bound on the size of the discretization of $O(1/\varepsilon)$, whereby the pseudo-dimension of this discretization can be bounded by the logarithm of the number of contracts in the discretization, i.e., $O(\ln(1/\varepsilon))$. The result then follows from their general framework, which only requires a bound on the pseudo-dimension of the contract class, and running the EUM algorithm on the discretization. Thus, the first bound does not provide a bound on the sample complexity of learning the class of linear contracts uniformly (all contracts simultaneously), as Theorem 1.1 does. Instead, it provides a bound on the sample complexity of learning a discretization of the class of linear contracts that preserves the optimal contract up to an additive error of $\varepsilon$, which is sufficient for the EUM algorithm to learn the optimal contract up to an additive error of $\varepsilon$ with probability at least $1 - \delta$. Furthermore, to the best of our knowledge this sample complexity bound combined with the EUM algorithm do as Algorithm 1, only need an oracle for the empirical utility over $\mathbf{S}$, and not the full information of the agents types.

The second bound is a bound on the pseudo-dimension of the class of linear contracts, which they show can be upper bounded by $O(\ln(n))$, this is done by relating the number of critical values of a linear contract (where the principal's expected reward changes), which is at most $n$, to the pseudo-dimension of the class of linear contracts.

This bound recovers the full generality of Theorem 1.1, but with the drawback of its sample complexity being dependent on the number of actions, $n$, which could be large and is assumed by the setting we consider to be unknown to the principal, thus, to the best of our knowledge to leverage the sample complexity bound of $O((\ln(n) + \ln(1/\delta))/\varepsilon^2)$, one would need to have knowledge of $n$ which is for instance provided in the basic setting of full information on the sample of (Dütting et al., 2025b).

Thus, in the regime of large $n > 1/\varepsilon$, there remained a gap between the best known sample complexity for learning the class of linear contracts uniformly and that of learning the best contract, which we close with the results, Theorem 1.1 and Theorem 1.2. Furthermore, Theorem 5.9 of (Dütting et al., 2025b), which shows a lower bound of $\Omega(\ln(1/\delta)/\varepsilon^2)$ on the sample complexity of learning the optimal linear contract up to $\varepsilon$-error, when combined with the results in Theorem 1.1 and Theorem 1.2, witnesses the tightness of all these results up to constant factors and shows that there is no gap in the sample complexity between learning the class of linear contracts uniformly and learning the best contract.

It is worth noting that the lower bound of (Dütting et al., 2025b) is for a simple distribution over two agent types with two actions, $n = 2$. Thus, the lower bound (and the previous uniform upper bound) did not rule out the possibility of the sample complexity being dependent on the number of actions, which we show is not the case.

In the remainder of the paper, we show how to prove the main results, Theorem 1.1 and Theorem 1.2. The proof uses a key structural property of the class of linear contracts, having a non-decreasing reward, highlighting how using specific properties of a contract space can be leveraged to obtain optimal sample complexity bounds. We hope this can lead to more optimal bounds in the area of contract design.

## 2. Proof Overview and Formal Proofs

In this section, we prove our main result Theorem 1.1 and its Theorem 1.2. We begin the section with a high-level overview of the proof of Theorem 1.1, and then show how it naturally implies Theorem 1.2, and finally, provide the full proof of Theorem 1.1.

The high-level proof idea of Theorem 1.1 is as follows: We first upper bound the difference between the empirical utility and the expected utility for all linear contracts, $\sup_{\alpha \in [0,1]} |u_p(\mathbf{S}, \alpha) - u_p(\mathcal{D}, \alpha)|$, by the Rademacher complexity, $\mathbb{E}_{\mathbf{S} \sim \mathcal{D}^s, \sigma \sim \{-1,1\}^s}[\sup_{\alpha \in [0,1]} \sum_{i=1}^s \sigma_i u_p(\theta_i, \alpha)/s]$, of the class of linear contracts plus $\varepsilon$, by McDiarmid's inequality as done in (Bartlett & Mendelson, 2003). This bound holds with probability at least $1 - \delta$ over $\mathbf{S} = (\theta_1, \ldots, \theta_s) \sim \mathcal{D}^s$, for $s = \Omega(\ln(1/\delta)/\varepsilon^2)$.

The upper bound on the generalization error in terms of the Rademacher complexity of linear contracts provides an intuitive explanation of the generalization property of linear contracts, as the Rademacher complexity measures how prone the linear contracts are to fitting random noise, i.e., if linear contracts could overfit to the data, leading to the empirical utility and expected utility being far from each other. However, as the argument shows, the linear contracts have a low Rademacher complexity, thus the empirical utility and expected utility are close to each other.

Having upper bounded the generalization error of linear contracts by its Rademacher complexity, we use a chaining result from (Rebeschini, 2021)[Proposition 5.3], to upper bound the Rademacher complexity of linear contracts in terms of their covering number, by the following relation:

$$\mathbb{E}_{\sigma \sim \{-1,1\}^s}\left[\sup_{\alpha \in [0,1]} \sum_{i=1}^s \sigma_i u_p(\theta_i, \alpha)/s\right] \quad (2)$$

$$\leq \inf_{\eta \in [0,1/2]}\left\{4\eta + \frac{12}{\sqrt{s}} \int_\eta^{1/2} \sqrt{\ln N(\mathcal{C}_{linear}, ||\cdot||_{2,\mathbf{S}}, \nu)} d\nu\right\}.$$

Here, $N(\mathcal{C}_{linear}, ||\cdot||_{2,\mathbf{S}}, \nu)$ is the covering number of linear contracts on the set of agents $\mathbf{S}$ with respect to the $L_2$ norm and precision $\nu$. This is the smallest integer such that there exists a set of linear contracts $\mathcal{C}_\nu$ of size $N(\mathcal{C}_{linear}, ||\cdot||_{2,\mathbf{S}}, \nu)$,

which satisfies that for any linear contract $\alpha \in \mathcal{C}_{linear}$, there exists $\widehat{\alpha} \in \mathcal{C}_\nu$ such that

$$\sqrt{\sum_{i=1}^{s}(u_p(\theta_i, \alpha) - u_p(\theta_i, \widehat{\alpha}))^2/s} \leq \nu.$$

We then show that if one can find a cover of size $O((1/\nu)^c)$ for some constant $c > 0$, Equation (2) reduces to $O(\inf_{\eta \in [0,1/2]}\{\eta + 1/\sqrt{s}\})$ which is $O(1/\sqrt{s})$. Since we set $s = \Omega(\ln(1/\delta)/\varepsilon^2)$ we have that $O(1/\sqrt{s}) = O(\varepsilon)$, which implies that the generalization error is $O(\varepsilon)$ with probability at least $1 - \delta$.

Thus, we have reduced the problem of bounding the generalization error of all linear contracts to bounding their covering number by $O((1/\nu)^c)$. We now proceed to show how to find such a cover. We will first find an $L_1$ cover of size $O(1/\nu)$ which, as we will show later, can be converted to an $L_2$ cover of size $O((1/\nu)^2)$, which by the above argument is sufficient to obtain the desired bound on the generalization error of linear contracts.

Now, an intuitive first approach one could explore to find such a cover would be to discretize the interval $[0, 1]$ into $O(1/\nu)$ points, being multiples of $\nu$, which would work if the utility of linear contracts were linear in the parameter $\alpha$. However, this is not the case, as the utility of linear contracts can have discontinuities and, in general, does not possess any monotonicity properties. See Figure 1 for an illustration. However, trying the above brings some insights that might be important to finding a small cover, namely if we let $\alpha \in [0, 1]$ and $\widehat{\alpha}$ be the point in the discretization of the interval $[0, 1]$ that is closest to $\alpha$, we have that

$$\frac{1}{s}\sum_{i=1}^{s}|u_p(\theta_i, \alpha) - u_p(\theta_i, \widehat{\alpha})| = \qquad (3)$$
$$\frac{1}{s}\sum_{i=1}^{s}\left|\sum_{j=1}^{m} f_{i^*(\theta_i, \alpha),j}(1 - \alpha)r_j - f_{i^*(\theta_i, \widehat{\alpha}),j}(1 - \widehat{\alpha})r_j\right|$$
$$\leq \frac{(1-\alpha)}{s}\sum_{i=1}^{s}\left|\sum_{j=1}^{m}\left(f_{i^*(\theta_i, \alpha),j}r_j - f_{i^*(\theta_i, \widehat{\alpha}),j}r_j\right)\right|$$
$$+ \underbrace{\frac{1}{s}\sum_{i=1}^{s}\left|\sum_{j=1}^{m} f_{i^*(\theta_i, \widehat{\alpha}),j}(\alpha - \widehat{\alpha})r_j\right|}_{\nu}$$

where the inequality follows from adding and subtracting $\alpha$ in the term $(1 - \widehat{\alpha})$ and using the triangle inequality. Now, since $|\alpha - \widehat{\alpha}| \leq \nu$, $r_j \in [0, 1]$, and $f_{i^*(\theta_i, \widehat{\alpha}),j}$ is a probability distribution, it follows from yet another use of the triangle inequality that the last term in the above is at most $\nu$. Thus, we have a bound on the second term of $\nu$, however we do not have control of the first term yet. Thus, we have to use a more refined approach. To this end, we use a result of (Dütting et al., 2021a), which shows that even though the empirical utility is not monotonic, the empirical reward $r_p(\theta, \alpha) := \sum_{j=1}^{m} f_{i^*(\theta, \alpha),j}r_j$ of linear contracts is a non-decreasing function in the contract parameter $\alpha \in [0, 1]$. See Figure 2 for an illustration. The insight is now that we can discretize the y-axis of the empirical reward

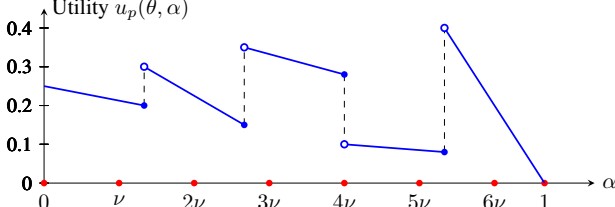

Figure 1. An example of the principal's utility $u_p(\theta, \alpha)$ as a function of the linear contract parameter $\alpha$. The utility can be non-monotonic and exhibit discontinuities. The red dots on the x-axis illustrate a simple discretization of the parameter space.

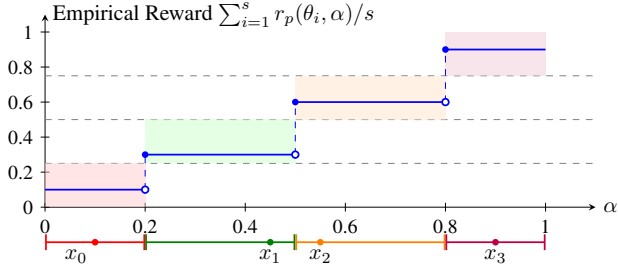

Figure 2. An example of the empirical reward $\sum_{i=1}^{s} r_p(\theta_i, \alpha)/s$ as a function of the linear contract parameter $\alpha$. The y-axis is discretized into intervals (separated by dashed lines). The pullback of these intervals onto the x-axis is shown as colored bars, and a point added to $\mathcal{C}_\nu$ from each pullback interval is marked, where in this example $x_0, x_1, x_2, x_3$ would be added to the discretization.

$\frac{1}{s}\sum_{i=1}^{s} r_p(\theta_i, \cdot)$ into intervals of length $O(\nu)$, take the pullback of each of these intervals, and add a point from each of these pullbacks to the discretization $\mathcal{C}_\nu$. Furthermore, we also discretize the x-axis into a grid of $O(1/\nu)$ equally spaced points and add these to $\mathcal{C}_\nu$. Now, for any linear contract $\alpha \in [0, 1]$, we have that $\frac{1}{s}\sum_{i=1}^{s} r_p(\theta_i, \alpha)$ takes a value in one of the intervals of length $\nu$, thus there exists a point $\widehat{\alpha} \in \mathcal{C}_\nu$ such that the pullback of the interval containing $\frac{1}{s}\sum_{i=1}^{s} r_p(\theta_i, \alpha)$ contains $\widehat{\alpha}$, and furthermore this point $\widehat{\alpha}$ can be chosen to be at most $\nu$ from the point $\alpha$. Now using the observation from the earlier attempt, Equation (3), we have that for such a contract $\widehat{\alpha} \in \mathcal{C}_\nu$,

$$\frac{1}{s}\sum_{i=1}^{s}|u_p(\theta_i, \alpha) - u_p(\theta_i, \widehat{\alpha})| =$$
$$\frac{1}{s}\sum_{i=1}^{s}\left|\sum_{j=1}^{m} f_{i^*(\theta_i, \alpha),j}(1 - \alpha)r_j - f_{i^*(\theta_i, \widehat{\alpha}),j}(1 - \widehat{\alpha})r_j\right|$$
$$\leq \frac{(1-\alpha)}{s}\sum_{i=1}^{s}\left|\sum_{j=1}^{m}\left(f_{i^*(\theta_i, \alpha),j}r_j - f_{i^*(\theta_i, \widehat{\alpha}),j}r_j\right)\right| + \nu$$

Furthermore, since the empirical reward is non-decreasing, and $\widehat{\alpha} \leq \alpha$ (or for $\alpha \leq \widehat{\alpha}$ with the order switched), implying that $\sum_{j=1}^{m}(f_{i^*(\theta_i, \alpha),j}r_j - f_{i^*(\theta_i, \widehat{\alpha}),j}r_j) \geq 0$, we can drop

the absolute value in the first term. Thus, we have that

$$(1-\alpha)\frac{1}{s}\sum_{i=1}^{s}\left|\sum_{j=1}^{m}\left(f_{i^*(\theta_i,\alpha),j}r_j - f_{i^*(\theta_i,\widehat{\alpha}),j}r_j\right)\right|$$

$$= (1-\alpha)\frac{1}{s}\sum_{i=1}^{s}\sum_{j=1}^{m}\left(f_{i^*(\theta_i,\alpha),j}r_j - f_{i^*(\theta_i,\widehat{\alpha}),j}r_j\right)$$

$$= (1-\alpha)(\frac{1}{s}\sum_{i=1}^{s}r_p(\theta_i,\alpha) - \frac{1}{s}\sum_{i=1}^{s}r_p(\theta_i,\widehat{\alpha}))$$

where we can now use that $\alpha$ and $\widehat{\alpha}$ were in the pullback of the same interval, so we have that $\frac{1}{s}\sum_{i=1}^{s}r_p(\theta_i,\alpha)$ is at most $\nu$ from $\frac{1}{s}\sum_{i=1}^{s}r_p(\theta_i,\widehat{\alpha})$, showing that

$$\sum_{i=1}^{s}|u_p(\theta_i,\alpha) - u_p(\theta_i,\widehat{\alpha})|/s \leq 2\nu.$$

Now, using that $|u_p(\theta_i,\alpha) - u_p(\theta_i,\widehat{\alpha})| \leq 1$, we conclude that

$$\sqrt{\sum_{i=1}^{s}(u_p(\theta_i,\alpha) - u_p(\theta_i,\widehat{\alpha}))^2/s} \leq \sqrt{2\nu}, \quad (4)$$

whereby rescaling $\nu$ to $\nu^2/2$ gives us the existence of a cover of size $O((1/\nu)^c)$ for the constant $c = 2$, which gives us the desired result.

With the high-level proof idea behind Theorem 1.1 explained, we now proceed to show how it implies the optimal sample complexity bound of the Empirical Utility Maximization (EUM) algorithm, i.e., Theorem 1.2.

## 2.1. Proof of Theorem 1.2

We now show that the simple Empirical Utility Maximization (EUM) algorithm over an appropriate set of linear contracts (the same as considered in (Dütting et al., 2025b)[Lemma 4.3]) gives an efficient algorithm for computing, with probability at least $1-\delta$, an $\varepsilon$ approximation of the best linear contract with the optimal sample complexity bound of Theorem 1.2. Formally, we consider the following algorithm.

---

**Algorithm 1** EUM for Linear Contracts

---

**Require:** An oracle for $u_p(\mathbf{S},\cdot)$ over a sample $\mathbf{S} = (\theta_1,\ldots,\theta_s)$, of size $s \geq 6912\ln(4/\delta)/\varepsilon^2$.
  Let $D_{\varepsilon/4} = \{0, \varepsilon/4, 2(\varepsilon/4), \ldots, \lfloor 4/\varepsilon\rfloor\varepsilon/4, 1\}$.
  **Return** $\widehat{\alpha^\star} \in \arg\max_{\alpha\in D_{\varepsilon/4}} u_p(\mathbf{S},\alpha)$.

---

Before we prove Theorem 1.2, we need the following lemma, which is due to (Dütting et al., 2021a). For completeness, we provide the proof in Section A.

**Lemma 2.1.** *The expected reward of linear contracts is non-decreasing in the contract parameter $\alpha \in [0,1]$, i.e. for any $\alpha' \geq \alpha$, it holds that*

$$\sum_{j=1}^{m}f_{a^\star(\theta,\alpha'),j}r_j = r_p(\theta,\alpha') \geq r_p(\theta,\alpha) = \sum_{j=1}^{m}f_{a^\star(\theta,\alpha),j}r_j.$$

Using Theorem 1.1 and Theorem 2.1, we now give the proof of Theorem 1.2, i.e., that Algorithm 1 obtains the optimal sample complexity for learning a linear contract that is $\varepsilon$-close to the optimal contract's utility.

*Proof of Theorem 1.2.* As noted in (Dütting et al., 2025b)[Lemma 4.3] we have that for $\alpha^\star \in [0,1]$ such that $u_p(\mathcal{D},\alpha^\star) = \sup_{\alpha\in[0,1]}u_p(\mathcal{D},\alpha) - \varepsilon/4$, it holds for the point $\alpha' \in D_{\varepsilon/4}$ that is closest to the right of $\alpha^\star$ that $u_p(\mathcal{D},\alpha') \geq u_p(\mathcal{D},\alpha^\star)$. This can be seen by the following calculation, using Theorem 2.1 and the fact that $\alpha'$ is the point in $D_{\varepsilon/4}$ closest to the right of $\alpha^\star$:

$$u_p(\mathcal{D},\alpha') = \mathbb{E}_{\theta\sim\mathcal{D}}\left[\sum_{j=1}^{m}f_{a^\star(\theta,\alpha'),j}(1-\alpha')r_j\right]$$

$$\geq \mathbb{E}_{\theta\sim\mathcal{D}}\left[\sum_{j=1}^{m}f_{a^\star(\theta,\alpha'),j}(1-\alpha^\star)r_j\right] - \varepsilon/4$$

$$\geq \mathbb{E}_{\theta\sim\mathcal{D}}\left[\sum_{j=1}^{m}f_{a^\star(\theta,\alpha^\star),j}(1-\alpha^\star)r_j\right] - \varepsilon/4$$

$$\geq \sup_{\alpha\in[0,1]}\mathbb{E}_{\theta\sim\mathcal{D}}\left[\sum_{j=1}^{m}f_{a^\star(\theta,\alpha),j}(1-\alpha^\star)r_j\right] - \varepsilon/2$$

where the first inequality follows from the fact that $\alpha'$ is the point in $D_{\varepsilon/4}$ closest to the right of $\alpha^\star$, the second inequality follows from Theorem 2.1 and $\alpha' \geq \alpha^\star$, so $\sum_{j=1}^{m}f_{a^\star(\theta,\alpha'),j}r_j \geq \sum_{j=1}^{m}f_{a^\star(\theta,\alpha^\star),j}r_j$, and the last inequality is due to the fact that $\alpha^\star$ is $\varepsilon/4$-close to the optimal utility $\sup_{\alpha\in[0,1]}\mathbb{E}_{\theta\sim\mathcal{D}}[\sum_{j=1}^{m}f_{a^\star(\theta,\alpha),j}(1-\alpha)r_j]$. Thus, we have that there in $D_{\varepsilon/4}$ exists an $\alpha'$ such that $u_p(\mathcal{D},\alpha') \geq \sup_{\alpha\in[0,1]}u_p(\mathcal{D},\alpha) - \varepsilon/2$. Now since $s = \lceil 13824\ln(4/\delta)/\varepsilon^2\rceil$, Theorem 1.1 implies that it holds with probability at least $1-\delta$, for all $\alpha \in [0,1]$, that

$$|u_p(S,\alpha) - u_p(\mathcal{D},\alpha)| \leq \varepsilon/2.$$

Thus, since $D_{\varepsilon/4} \subseteq \mathcal{C}_{linear} = [0,1]$, the above event also holds for all the contracts in $D_{\varepsilon/4}$ with probability at least $1-\delta$. Thus, we have that with probability at least $1-\delta$, it holds for $\widehat{\alpha^\star}$ that

$$u_p(S,\widehat{\alpha^\star}) \geq \sup_{\alpha\in D_{\varepsilon/4}}u_p(S,\alpha)$$

$$\geq \sup_{\alpha\in D_{\varepsilon/4}}u_p(\mathcal{D},\alpha) - \varepsilon/2 \geq \sup_{\alpha\in[0,1]}u_p(\mathcal{D},\alpha) - \varepsilon,$$

where we in the first inequality used that $\widehat{\alpha^\star}$ is the maximizer of the empirical utility over $D_{\varepsilon/4}$, in the second inequality we used that the empirical utility is close to the expected utility for all linear contracts in $D_{\varepsilon/4}$ with probability at least $1-\delta$, and the last inequality follows from the fact that, as argued above, there exists an $\alpha'$ in $D_{\varepsilon/4}$ such that $u_p(\mathcal{D},\alpha') \geq \sup_{\alpha\in[0,1]}u_p(\mathcal{D},\alpha)-\varepsilon/2$. Thus, we have that with probability at least $1-\delta$, it holds that

$$u_p(\mathcal{D},\widehat{\alpha^\star}) \geq \sup_{\alpha\in[0,1]}u_p(\mathcal{D},\alpha) - \varepsilon.$$

We furthermore notice that since $D_{\varepsilon/4}$ contains at most $\lfloor 4/\varepsilon\rfloor + 2 \leq 6/\varepsilon$ contracts, the algorithm Algorithm 1 only has to query the oracle for $u_p(\mathbf{S},\cdot)$ at most $O(\frac{1}{\varepsilon})$-times, as claimed in Theorem 1.2, which concludes the proof. $\square$

## 2.2. Proof of Theorem 1.1

We now proceed to give the proof of Theorem 1.1. To show Theorem 1.1, we will use the following lemma giving a bound on the $L_2$ cover of linear contracts of size $O(1/\nu^2)$, which is the main technical contribution.

**Lemma 2.2.** *For any $1 > \nu > 0$ and $S = (\theta_1, \ldots, \theta_s)$, there exists a set of linear contracts $\mathcal{C}_\nu \subset [0, 1]$ such that*

- $|\mathcal{C}_\nu| = 12/\nu^2$
- *For any linear contract $\alpha \in [0, 1]$, there exists a contract $\widehat{\alpha} \in \mathcal{C}_\nu$ such that*

$$\sqrt{\sum_{i=1}^s |u_p(\theta_i, \alpha) - u_p(\theta_i, \widehat{\alpha})|^2 / s} \le \nu.$$

With Theorem 2.2 in hand, we can now prove Theorem 1.1.

*Proof of Theorem 1.1.* To show Theorem 1.1, we consider the random variable $\sup_{\alpha \in [0,1]} u_p(\mathbf{S}, \alpha) - u_p(\mathcal{D}, \alpha)$ (and $\sup_{\alpha \in [0,1]} u_p(\mathcal{D}, \alpha) - u_p(\mathbf{S}, \alpha)$). Notice that, by $r_j \in [0, 1]$, we have that $(u_p(\mathcal{D}, \alpha) - u_p(\theta, \alpha))/s \in [-1/s, 1/s]$ for any $\theta \in S$. Thus, by McDiarmid's inequality we obtain that with probability at least $1 - 2\exp(\varepsilon^2 s/8)$, it holds that

$$\sup_{\alpha \in [0,1]} u_p(\mathbf{S}, \alpha) - u_p(\mathcal{D}, \alpha) \qquad (5)$$
$$\in \mathop{\mathbb{E}}_{\mathbf{S} \sim \mathcal{D}^s} \left[ \sup_{\alpha \in [0,1]} u_p(\mathbf{S}, \alpha) - u_p(\mathcal{D}, \alpha) \right] \pm \varepsilon/2.$$

In order to control the above expectation term, we make the following calculation, starting with a symmetrization step. To this end, let $\mathbf{S}' = (\theta'_1, \ldots, \theta'_s) \sim \mathcal{D}^s$. We then get that

$$\mathop{\mathbb{E}}_{\mathbf{S} \sim \mathcal{D}^s} \left[ \sup_{\alpha \in [0,1]} u_p(\mathbf{S}, \alpha) - u_p(\mathcal{D}, \alpha) \right] \qquad (6)$$
$$= \mathop{\mathbb{E}}_{\mathbf{S} \sim \mathcal{D}^s} \left[ \sup_{\alpha \in [0,1]} \mathbb{E}_{\mathbf{S}' \sim \mathcal{D}^s} \left[ u_p(\mathbf{S}, \alpha) - u_p(\mathbf{S}', \alpha) \right] \right]$$
$$\le \mathop{\mathbb{E}}_{\mathbf{S} \sim \mathcal{D}^s} \left[ \mathop{\mathbb{E}}_{\mathbf{S}' \sim \mathcal{D}^s} \left[ \sup_{\alpha \in [0,1]} u_p(\mathbf{S}, \alpha) - u_p(\mathbf{S}', \alpha) \right] \right]$$
$$= \mathop{\mathbb{E}}_{\mathbf{S}, \mathbf{S}' \sim \mathcal{D}^s} \left[ \sup_{\alpha \in [0,1]} \sum_{i=1}^s \left( u_p(\theta_i, \alpha) - u_p(\theta'_i, \alpha) \right)/s \right]$$
$$= \mathop{\mathbb{E}}_{\mathbf{S}, \mathbf{S}' \sim \mathcal{D}^s, \sigma} \left[ \sup_{\alpha \in [0,1]} \sum_{i=1}^s \sigma_i \left( u_p(\theta_i, \alpha) - u_p(\theta'_i, \alpha) \right)/s \right]$$
$$\le 2 \mathop{\mathbb{E}}_{\mathbf{S} \sim \mathcal{D}^s, \sigma \sim \{-1,1\}^s} \left[ \sup_{\alpha \in [0,1]} \sum_{i=1}^s \sigma_i u_p(\theta_i, \alpha)/s \right]$$

where the first inequality follows from the fact that taking sup inside the expectation only increases the expectation; in the last equality, we used the i.i.d. assumption of the samples $\mathbf{S}$ and $\mathbf{S}'$, meaning that $u_p(\theta_i, \alpha) - u_p(\theta'_i, \alpha)$ has the same distribution as $u_p(\theta'_i, \alpha) - u_p(\theta_i, \alpha)$ for $i \in [s]$ (and each term being independent); and the last inequality follows from sup of the difference being less than the sum of the sup of each term and that $-\sigma_i u_p(\theta'_i, \alpha)$ has the same distribution as $\sigma_i u_p(\theta_i, \alpha)$ for $i \in [s]$ (and each term being independent). For any realization $S$ of $\mathbf{S}$, we have from,

e.g., (Rebeschini, 2021)[Proposition 5.3], that

$$\mathop{\mathbb{E}}_{\sigma \sim \{-1,1\}^s} \left[ \sup_{\alpha \in [0,1]} \sum_{i=1}^s \sigma_i u_p(\theta_i, \alpha)/s \right] \qquad (7)$$
$$\le \inf_{\eta \in [0,1/2]} \left\{ 4\eta + \frac{12}{\sqrt{s}} \int_\eta^{1/2} \sqrt{\ln\left(N(\mathcal{C}_{linear}, ||\cdot||_{2,S}, \nu)\right)} d\nu \right\},$$

where $N(\mathcal{C}_{linear}, ||\cdot||_{2,S}, \nu)$ is the covering number of $\mathcal{C}_{linear} = [0, 1]$ on $S$ with respect to the $L_2$ norm and precision $\nu$, i.e., the smallest number such that there exists a set of contracts $\mathcal{C}_\nu$ of size $N(\mathcal{C}_{linear}, ||\cdot||_{2,S}, \nu)$ such that for any $\alpha \in [0, 1]$, there exists $\widehat{\alpha} \in \mathcal{C}_\nu$ such that $\sqrt{\sum_{i=1}^s |u_p(\theta_i, \alpha) - u_p(\theta_i, \widehat{\alpha})|^2 / s} \le \nu$. We notice that if we can show that $N(\mathcal{C}_{linear}, ||\cdot||_{2,S}, \nu) \le 12/\nu^2$ (which is exactly what Theorem 2.2 implies), we obtain by Equation (7) that

$$\mathop{\mathbb{E}}_{\sigma \sim \{-1,1\}^s} \left[ \sup_{\alpha \in [0,1]} \sum_{i=1}^s \sigma_i u_p(\theta_i, \alpha)/s \right] \qquad (8)$$
$$\le \inf_{\eta \in (0,1/2]} \left\{ 4\eta + \frac{12}{\sqrt{s}} \int_\eta^{1/2} \sqrt{\ln\left(N([0,1], ||\cdot||_{2,S}, \nu)\right)} d\nu \right\}$$
$$\le \inf_{\eta \in (0,1/2]} \left\{ 4\eta + \frac{12}{\sqrt{s}} \int_\eta^{1/2} \sqrt{\ln\left(12/\nu^2\right)} d\nu \right\}$$
$$= \inf_{\eta \in (0,1/2]} \left\{ 4\eta + \frac{12\sqrt{2}}{\sqrt{s}} \int_\eta^{1/2} \sqrt{\ln\left(\sqrt{12}/\nu\right)} d\nu \right\}$$
$$= \inf_{\eta \in (0,1/2]} \left\{ 4\eta + \frac{12 \cdot \sqrt{2 \cdot 12}}{\sqrt{s}} \int_{\eta/\sqrt{12}}^{1/(2 \cdot \sqrt{12})} \sqrt{\ln\left(1/\nu'\right)} d\nu' \right\}$$
$$\le \inf_{\eta \in (0,1/2]} \left\{ 4\eta + \frac{12 \cdot \sqrt{2 \cdot 12}}{\sqrt{s}} \int_0^{1/(2 \cdot \sqrt{12})} \sqrt{\ln\left(1/\nu'\right)} d\nu' \right\}$$
$$\le \inf_{\eta \in (0,1/2]} \left\{ 4\eta + \frac{12 \cdot \sqrt{2 \cdot 12}}{\sqrt{s}} \frac{1}{4} \right\} = \frac{6 \cdot \sqrt{6}}{\sqrt{s}}$$

the first equality follows from integration by substitution $\nu' = \nu/\sqrt{12}$, the third to last inequality follows from that $\int_0^{1/(2\cdot\sqrt{12})} \sqrt{\ln(1/\nu')} d\nu' \le 1/4$, and the last equality follows from $\inf_{\eta \in (0,1/2]}$ making $4\eta$ vanish. We note that we showed the above for any realization $S$ of $\mathbf{S}$, thus it also holds for random $\mathbf{S}$. Now, combining the conclusion of Equation (6) and Equation (8) we have shown that

$$\mathop{\mathbb{E}}_{\mathbf{S} \sim \mathcal{D}^s} \left[ \sup_{\alpha \in [0,1]} u_p(\mathbf{S}, \alpha) - u_p(\mathcal{D}, \alpha) \right] \le \frac{12 \cdot \sqrt{6}}{\sqrt{s}} \qquad (9)$$

To the end of using the conclusion of Equation (5) and Equation (9), we set $s \ge (2 \cdot 12 \cdot \sqrt{6})^2 \ln(4/\delta)/\varepsilon^2$. Then by Equation (5), we have that with probability at least $1 - \delta/2$ over $\mathbf{S}$ that

$$\sup_{\alpha \in [0,1]} u_p(\mathbf{S}, \alpha) - u_p(\mathcal{D}, \alpha)$$
$$\in \mathop{\mathbb{E}}_{\mathbf{S} \sim \mathcal{D}^s} \left[ \sup_{\alpha \in [0,1]} u_p(\mathbf{S}, \alpha) - u_p(\mathcal{D}, \alpha) \right] \pm \varepsilon/2$$

and by Equation (9) that

$$0 \le \mathop{\mathbb{E}}_{\mathbf{S} \sim \mathcal{D}^s} \left[ \sup_{\alpha \in [0,1]} u_p(\mathbf{S}, \alpha) - u_p(\mathcal{D}, \alpha) \right] \le \varepsilon/2,$$

where the lower bound follows by for $\alpha = 1$ the $u_p(\mathbf{S}, 1), u_p(\mathcal{D}, 1) = 0$. Thus we have shown that with probability at least $1 - \delta/2$ over $\mathbf{S}$ that

$$-\varepsilon \leq \sup_{\alpha \in [0,1]} u_p(\mathbf{S}, \alpha) - u_p(\mathcal{D}, \alpha) \leq \varepsilon.$$

Now, repeating the above argument with $\sup_{\alpha \in [0,1]} u_p(\mathcal{D}, \alpha) - u_p(\mathbf{S}, \alpha)$ gives that with probability at least $1 - \delta/2$,

$$-\varepsilon \leq \sup_{\alpha \in [0,1]} u_p(\mathcal{D}, \alpha) - u_p(\mathbf{S}, \alpha) \leq \varepsilon,$$

which by the union bound concludes the proof. □

With the proof of Theorem 1.1 given using Theorem 2.2, we now proceed to give the proof of Theorem 2.2.

*Proof of Theorem 2.2.* We first prove an auxiliary $L_1$ cover statement. Namely, we show that for any set of agents $\theta_1, \ldots, \theta_s$, reward vector $r \in [0, 1]^m$, and precision parameter $0 < \nu' < 1$, there exists a set of contracts $\widetilde{\mathcal{C}}_{\nu'} \subseteq [0, 1]$ of size $|\widetilde{\mathcal{C}}_{\nu'}| \leq 12/\nu'$ such that for any $\alpha \in [0, 1]$, there exists $\widehat{\alpha} \in \widetilde{\mathcal{C}}_{\nu'}$ for which it holds that

$$\sum_{i=1}^{s} |u_p(\theta_i, \alpha) - u_p(\theta_i, \widehat{\alpha})|/s \leq \nu'. \quad (10)$$

We now explain why this auxiliary statement implies Theorem 2.2. Fix $0 < \nu < 1$. Since $u_p(\theta_i, \alpha) = \sum_{j=1}^{m} f_{a^\star(\theta_i, \alpha), j}(1 - \alpha) r_j \in [0, 1]$ for all $\alpha \in [0, 1]$, applying Equation (10) with precision $\nu' = \nu^2$ gives a set $\widetilde{\mathcal{C}}_{\nu^2}$ of size at most $12/\nu^2$ such that for any $\alpha \in [0, 1]$, there exists $\widehat{\alpha} \in \widetilde{\mathcal{C}}_{\nu^2}$ satisfying

$$\sqrt{\sum_{i=1}^{s} |u_p(\theta_i, \alpha) - u_p(\theta_i, \widehat{\alpha})|^2/s}$$
$$\leq \sqrt{\sum_{i=1}^{s} |u_p(\theta_i, \alpha) - u_p(\theta_i, \widehat{\alpha})|/s} \leq \sqrt{\nu^2} = \nu,$$

where the first inequality follows from $|u_p(\theta_i, \alpha) - u_p(\theta_i, \widehat{\alpha})| \leq 1$ and the second follows from Equation (10) with precision $\nu' = \nu^2$. Thus $\widetilde{\mathcal{C}}_{\nu^2}$ is an $L_2$ cover at precision $\nu$ with size at most $12/\nu^2$; renaming this set to $\mathcal{C}_\nu$ establishes Theorem 2.2. It remains to prove the auxiliary statement Equation (10), to keep notation light we will use $\nu$ instead of $\nu'$. To this end, consider any sequence $\theta_1, \ldots, \theta_s$ of agents and let $0 < \nu < 1$.

The empirical reward of a linear contract $t = \alpha r$ can be written as $\sum_{i=1}^{s} r_p(\theta_i, \alpha)/s$. By Theorem 2.1, we know that for each $\theta_i$, $r_p(\theta_i, \alpha)$ is a non-decreasing function in $\alpha$. Thus, we also have that the empirical reward $\sum_{i=1}^{s} r_p(\theta_i, \alpha)/s$ is a non-decreasing function in $\alpha$. We now consider two discretizations, $\widetilde{\mathcal{C}}_1$ and $\widetilde{\mathcal{C}}_2$, of the interval $[0, 1]$.

We first discretize the interval $[0, 1]$ into points $x_{1,i} = i\nu$, for $i \in I = \{0, \ldots, \lfloor 1/\nu \rfloor, \lfloor 1/\nu \rfloor + 1\}$, with $x_{1, \lfloor 1/\nu \rfloor + 1} =$

1, and set $\widetilde{\mathcal{C}}_1 = \cup_{i \in I} x_{1,i}$. Thus, for any $\alpha \in [0, 1]$, there is a point in $\widetilde{\mathcal{C}}_1$ which is at most $\nu$-close to $\alpha$. We now "discretize" the y-axis and take the pullback of $\sum_{i=1}^{s} r_p(\theta_i, \alpha)/s$, such that the pullback of the values forms a net for $\sum_{i=1}^{s} r_p(\theta_i, \alpha)/s$.

Formally, let $y_i = i\nu$ for $i \in I = \{0, \ldots, \lfloor 1/\nu \rfloor, \lfloor 1/\nu \rfloor + 1\}$, with $y_{\lfloor 1/\nu \rfloor + 1} = 1$. For each $i \in I$, if there exists a value $x' \in [0, 1]$ such that

$$y_i \leq \sum_{k=1}^{s} r_p(\theta_k, x')/s < y_{i+1}, \quad (11)$$

let $x_{2,i}$ be any such $x'$ (else skip this value) (where $y_{\lfloor 1/\nu \rfloor + 2} = 1 + \nu$). Furthermore, let $X_i = \{x \in [0, 1] : y_i \leq \sum_{k=1}^{s} r_p(\theta_k, x)/s < y_{i+1}\}$. We notice that since $\sum_{k=1}^{s} r_p(\theta_k, \alpha)/s$ is a non-decreasing function in $\alpha$, $X_i$ is an interval. Let $\widetilde{\mathcal{C}}_2 = \cup_{i \in I} x_{2,i}$.

Set the final discretization equal to $\widetilde{\mathcal{C}}_\nu = \widetilde{\mathcal{C}}_1 \cup \widetilde{\mathcal{C}}_2$. We notice that by the above construction, we have that $|\widetilde{\mathcal{C}}_\nu| \leq 2|I| \leq 2(\lfloor 1/\nu \rfloor + 2) \leq 6/\nu$.

Now consider any $\alpha \in [0, 1]$. Now, since $\sum_{i=1}^{s} r_p(\theta_i, \alpha)/s \in [0, 1]$ and $\cup_{i \in I} [y_i, y_{i+1}) = [0, 1 + \nu)$, it must be the case that there exists a $j \in I$ such that $y_j \leq \sum_{i=1}^{s} r_p(\theta_i, \alpha)/s \in [0, 1] < y_{j+1}$, where $y_{\lfloor 1/\nu \rfloor + 2} = 1 + \nu$, consider this $j$ for now. By the above construction of $\widetilde{\mathcal{C}}_2$, we have that there exists $x_{2,j} \in \widetilde{\mathcal{C}}_2$ such that $y_j \leq \sum_{i=1}^{s} r_p(\theta_i, x_{2,j})/s < y_{j+1}$, implying that $\widetilde{\mathcal{C}}_\nu \cap X_j \neq \emptyset$. Now let $\hat{\alpha}$ be the point closest to $\alpha$ in $\widetilde{\mathcal{C}}_\nu \cap X_j$. We observe that if $\hat{\alpha} \leq \alpha$, then

$$y_j \leq \sum_{i=1}^{s} r_p(\theta_i, \widehat{\alpha})/s \leq \sum_{i=1}^{s} r_p(\theta_i, \alpha)/s < y_{j+1}, \quad (12)$$

where the first inequality follows by definition of $\widehat{\alpha} \in X_j$, the second inequality follows from $\hat{\alpha} \leq \alpha$ and Theorem 2.1, and the last inequality follows by $\alpha \in X_j$. We notice that Theorem 2.1 and $\widehat{\alpha} \leq \alpha$ imply that $0 \leq r_p(\theta_i, \alpha)/s - r_p(\theta_i, \widehat{\alpha})/s = |r_p(\theta_i, \alpha)/s - r_p(\theta_i, \widehat{\alpha})/s|$. This, combined with Equation (12) implies that

$$0 \leq \sum_{i=1}^{s} |r_p(\theta_i, \alpha)/s - r_p(\theta_i, \widehat{\alpha})/s| = \quad (13)$$
$$\sum_{i=1}^{s} r_p(\theta_i, \alpha)/s - \sum_{i=1}^{s} r_p(\theta_i, \widehat{\alpha})/s \leq y_{j+1} - y_j \leq \nu.$$

In the case that $\hat{\alpha} > \alpha$, we have by a similar argument that

$$0 \leq \sum_{i=1}^{s} |r_p(\theta_i, \alpha)/s - r_p(\theta_i, \widehat{\alpha})/s| = \quad (14)$$
$$\sum_{i=1}^{s} r_p(\theta_i, \widehat{\alpha})/s - \sum_{i=1}^{s} r_p(\theta_i, \alpha)/s \leq y_{j+1} - y_j \leq \nu.$$

We furthermore observe that $|\hat{\alpha} - \alpha| \leq \nu$, since $X_j$ is an interval, so if it does not contain a point in $\widetilde{\mathcal{C}}_1$, it must have length strictly less than $\nu$, by the points in $\widetilde{\mathcal{C}}_1$ being at most $\nu$ from each other. In this case, we have that the point $\hat{\alpha}$ in $\widetilde{\mathcal{C}}_2$ from $X_j$ is at most $\nu$ away from $\alpha$. Otherwise, $X_j$

contains a point in $\widetilde{\mathcal{C}}_1$, and thus $\hat{\alpha}$ is at most $\nu$ away from $\alpha$, as we choose it as the closest point to $\alpha$ among the points in $\widetilde{\mathcal{C}}_\nu \cap X_j$. Now, using the above observations, we conclude that

$$
\begin{aligned}
&\sum_{i=1}^s |u_p(\theta_i, \alpha) - u_p(\theta_i, \widehat{\alpha})|/s \\
&= \sum_{i=1}^s | \sum_{j=1}^m f_{a^\star(\theta_i,\alpha),j}(1 - \alpha)r_j \\
&\quad - \sum_{j=1}^m f_{a^\star(\theta_i,\widehat{\alpha}),j}(1 - \widehat{\alpha})r_j|/s \\
&\le \sum_{i=1}^s | \sum_{j=1}^m (f_{a^\star(\theta_i,\alpha),j} - f_{a^\star(\theta_i,\widehat{\alpha}),j})(1 - \alpha)r_j|/s \\
&\quad + \sum_{i=1}^s | \sum_{j=1}^m f_{a^\star(\theta_i,\widehat{\alpha}),j}(\alpha - \widehat{\alpha})r_j|/s \\
&\le (1 - \alpha) \sum_{i=1}^s |r_p(\theta_i, \alpha) - r_p(\theta_i, \widehat{\alpha})|/s \\
&\quad + \sum_{i=1}^s \sum_{j=1}^m f_{a^\star(\theta_i,\widehat{\alpha}),j}|(\alpha - \widehat{\alpha})|r_j/s \le 2\nu,
\end{aligned}
$$

where the first equality follows from the definition of the principal's utility, the first inequality follows from the triangle inequality, the second inequality uses in the first sum the definition of the reward of the principal and in the second sum the triangle inequality $m$ times, and the third inequality uses in the first sum Equation (13) or Equation (14) and in the second sum that $|\alpha - \widehat{\alpha}| \le \nu, |r_j| \le 1$, and that $f_{a^\star(\theta_i,\widehat{\alpha})}$ forms a probability distribution. Thus, we have shown that $\widetilde{\mathcal{C}}_\nu$ is a cover for the linear contracts $[0, 1]$ on the agents $\theta_1, \ldots, \theta_s$, in $L_1$, with precision $2\nu$, and size $|\widetilde{\mathcal{C}}_\nu| \le 6/\nu$. Rescaling $\nu$ to $\nu/2$ concludes the proof of Equation (10), the claim of the size above Equation (10), and concludes the proof of Theorem 2.2. □

## 3. Discussion

Beyond linear contracts, our proof suggests a general route for obtaining sharp sample complexity bounds. The chaining argument itself is a standard tool from learning theory: once sufficiently small empirical $L_2$ covers are available for the utility class induced by a contract class, it yields generalization bounds through the corresponding covering numbers. Thus, the main contract-specific challenge is not the chaining step, but rather constructing, or proving the existence of, small covers that exploit the structure of the contract class.

In the case of linear contracts, directly constructing the cover from the monotonicity of the principal's expected reward removes the extra logarithmic or action-dependent terms appearing in previous bounds. It would be interesting to understand whether similar direct-cover arguments can give tighter, and possibly optimal, sample complexity bounds for other contract classes. Another natural direction is to determine whether, for richer contract classes, there can be a genuine sample complexity gap between uniform convergence over the class and merely learning the best contract in the class.

## Acknowledgements

While this work was carried out, Mikael Møller Høgsgaard was supported by an Internationalisation Fellowship from the Carlsberg Foundation. Furthermore, Mikael Møller Høgsgaard was supported by the European Union (ERC, TUCLA, 101125203). Views and opinions expressed are however those of the author(s) only and do not necessarily reflect those of the European Union or the European Research Council. Neither the European Union nor the granting authority can be held responsible for them. Lastly, Mikael Møller Høgsgaard was also supported by Independent Research Fund Denmark (DFF) Sapere Aude Research Leader grant No. 9064-00068B.

## Impact Statement

This paper presents work whose goal is to advance the field of Machine Learning. There are many potential societal consequences of our work, none which we feel must be specifically highlighted here.

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

# A. Appendix

In this appendix, we prove Theorem 2.1, which we restate here for convenience.

**Lemma A.1.** *The expected reward of linear contracts is non-decreasing in the contract parameter $\alpha \in [0, 1]$, i.e., for any $\alpha' > \alpha$ it holds that*

$$\sum_{j=1}^{m} f_{a^\star(\theta,\alpha'),j} r_j = r_p(\theta, \alpha') \geq r_p(\theta, \alpha) = \sum_{j=1}^{m} f_{a^\star(\theta,\alpha),j} r_j.$$

*Proof of Theorem 2.1.* We first recall that the utility of a linear contract $\alpha$ for an agent is given by

$$\sum_{j=1}^{m} f_{a^\star(\theta,\alpha),j} \alpha r_j - c_{a^\star(\theta,\alpha)},$$

where $a^\star(\theta, \alpha)$ is chosen as the action that maximizes the $\sum_{j=1}^{m} f_{i,j} \alpha r_j - c_i$, over $i$. Let $\alpha' > \alpha$. We then have that if the agent is offered the contract with parameter $\alpha'$, then the agent will choose the action $a^\star(\theta, \alpha')$ which maximizes her utility, i.e.,

$$\sum_{j=1}^{m} f_{a^\star(\theta,\alpha'),j} \alpha' r_j - c_{a^\star(\theta,\alpha')} \geq \sum_{j=1}^{m} f_{a^\star(\theta,\alpha),j} \alpha' r_j - c_{a^\star(\theta,\alpha)}$$

$$\Rightarrow \sum_{j=1}^{m} f_{a^\star(\theta,\alpha'),j} \alpha' r_j - c_{a^\star(\theta,\alpha')} - \left( \sum_{j=1}^{m} f_{a^\star(\theta,\alpha),j} \alpha' r_j - c_{a^\star(\theta,\alpha)} \right) \geq 0.$$

Furthermore, if the agent is offered the contract with parameter $\alpha$, then she will choose the action $a^\star(\theta, \alpha)$ which maximizes her utility, i.e.,

$$\sum_{j=1}^{m} f_{a^\star(\theta,\alpha),j} \alpha r_j - c_{a^\star(\theta,\alpha)} \geq \sum_{j=1}^{m} f_{a^\star(\theta,\alpha'),j} \alpha r_j - c_{a^\star(\theta,\alpha')}$$

$$\Rightarrow \sum_{j=1}^{m} f_{a^\star(\theta,\alpha),j} \alpha r_j - c_{a^\star(\theta,\alpha)} - \left( \sum_{j=1}^{m} f_{a^\star(\theta,\alpha'),j} \alpha r_j - c_{a^\star(\theta,\alpha')} \right) \geq 0.$$

Adding the latter two inequalities in the two above equations we get that

$$0 \leq \sum_{j=1}^{m} f_{a^\star(\theta,\alpha'),j} \alpha' r_j - c_{a^\star(\theta,\alpha')} - \left( \sum_{j=1}^{m} f_{a^\star(\theta,\alpha),j} \alpha' r_j - c_{a^\star(\theta,\alpha)} \right)$$

$$+ \sum_{j=1}^{m} f_{a^\star(\theta,\alpha),j} \alpha r_j - c_{a^\star(\theta,\alpha)} - \left( \sum_{j=1}^{m} f_{a^\star(\theta,\alpha'),j} \alpha r_j - c_{a^\star(\theta,\alpha')} \right)$$

$$= \sum_{j=1}^{m} f_{a^\star(\theta,\alpha'),j} (\alpha' - \alpha) r_j + \sum_{j=1}^{m} f_{a^\star(\theta,\alpha),j} (\alpha - \alpha') r_j$$

$$\underset{\text{dividing with } \alpha' - \alpha > 0}{\Rightarrow} \quad 0 \leq \sum_{j=1}^{m} f_{a^\star(\theta,\alpha'),j} r_j - \sum_{j=1}^{m} f_{a^\star(\theta,\alpha),j} r_j,$$

implying that

$$\sum_{j=1}^{m} f_{a^\star(\theta,\alpha'),j} r_j \geq \sum_{j=1}^{m} f_{a^\star(\theta,\alpha),j} r_j.$$

which, since $\alpha' > \alpha$, shows that the expected reward is non-decreasing in the contract parameter $\alpha$, and concludes the proof of Theorem 2.1. $\qquad\square$

# B. Definition of Pseudo-Dimension of Contracts

In this section, we restate the definition of pseudo-dimension from (Dütting et al., 2025b) for easy lookup.

**Definition B.1** (Pseudo-Dimension of Contracts (Dütting et al., 2025b)[Definition 3.5)**.** ] The pseudo-dimension of a contract class $\mathcal{C}$ with respect to an agent type space $\Theta$ is the largest integer $d$ such that there exists a set of agents $\theta_1, \ldots, \theta_d \in \Theta$ and real numbers $z_1, \ldots, z_d \in \mathbb{R}$ such that for any binary vector $b \in \{0, 1\}^d$, there exists a contract $t_b \in \mathcal{C}$ such that for all $i \in [d]$, it holds that

$$u_p(\theta_i, t_b) \geq z_i \text{ if } b_i = 1, \text{ and } u_p(\theta_i, t_b) < z_i \text{ if } b_i = 0.$$

If no such largest integer exists, then the pseudo-dimension is infinite.

