# OpenReview forum: "The Optimal Sample Complexity of Linear Contracts"
_ICML.cc/2026/Conference — ICML 2026 regular_

### Official Review · Reviewer_DA5P · 2026-02-15

**Soundness:** 3
**Presentation:** 2
**Significance:** 2
**Originality:** 2
**Overall Recommendation:** 3
**Confidence:** 4

**Summary:**

This paper studies the problem to learn an optimal linear contract in the principal-agent contract design problem, coming up with the result that empirical utility maximization yields an $\epsilon$-approximately optimal linear contract within $\tilde{O}(1/\epsilon^2)$ samples.

**Compliance With Llm Reviewing Policy:**

Affirmed.

**Final Justification:**

Based on the rebuttal, a significant amount of revision still need to be done before the paper becomes well prepared. So I would like to raise the score to weak reject but no further.

**Key Questions For Authors:**

--Is there any result on approximate optimality of linear contract? For example, an optimal linear contract yields a fraction of the optimal utility among all contracts, at least under some assumptions? It would make this paper more valuable in the field of learning-based contract design.

--Or is there any justification that linear contracts are somehow better than other contracts, so that it is worth to sacrifice some utility?

--In real-world applications, the learning-based contract design task may often run in an online setting. Is there any comparable results in the online setting? It would be more interesting.

**Limitations:**

The authors did not discuss about the limitations of only considering linear contracts.

I don't think there is any significant negative societal impact of the work.

**Strengths And Weaknesses:**

Strength:

-- The authors researches the contract design problem from a learning perspective, relaxing the unrealistic assumption that the types are known.

Weakness:

-- The authors claim that the result matches a lower bound in (D¨utting et al., 2025). However, the result of this paper is about *sample complexity* (how accurate we can learn from the data even given unbounded computation) while the lower bound in (D¨utting et al., 2025) is among PTAS algorithms (assuming computationally bounded principals) and about *computational complexity*. Hence, the two bounds may not be directly comparable.

-- The motivation and significance of learning an optimal *linear* contract is unclear, as the family of linear contracts lie in a small subset of all contracts. Actually in a learning setting, if we have a large dataset, why not just learn an *optimal contract* which may have much better utility?

-- The methodology of empirical utility maximization is straightforward and the result is not surprising, particularly for the space of linear contracts. The space of all linear contracts is only 1-dimensional. Learning an $\epsilon$-approximation of an 1-dimensional parameter with $\tilde{O}(1/\epsilon^2)$ samples is not a challenging task.

---

> ### Author Rebuttal · Authors · 2026-03-30
>
> Dear Reviewer DA5P,
>
> Thank you for reading and assessing the paper carefully. We will now address the reviewer's questions (Questions shortened due to the 5000-character limit):
>
> **Question 1**
> >"The authors claim that the result matches a lower bound in (Dütting et al., 2025) ... the two bounds may not be directly comparable."
>
> **Answer 1:**
>
> We apologize for the confusion caused by our bibliography, which contains two 2025 papers by Dütting et al. The reviewer is entirely correct that [1] focuses on computational aspects and does not contain these bounds. However, the lower bound is found in [2], which is the paper we cite for these results. We will clearly distinguish these two references in the next version to prevent any future ambiguity.
>
> **Question 2:**
> >"The motivation and significance of learning an optimal linear ... have much better utility?"
>
> > "Is there any result on approximate optimality of linear contract? ... learning-based contract design."
>
> >"Or is there any justification ... sacrifice some utility?"
>
> **Answer 2:**
>
> There are many reasons why linear contracts are interesting. First, their simplicity allows us to obtain non-vacuous generalization bounds at sample sizes where bounds for more advanced contracts remain vacuous. Thus, although complex contracts might achieve better empirical utility, we cannot be confident they will generalize to new samples, and using them may therefore lead to a loss in utility.
>
> From a computational point of view, linear contracts are more tractable to find. Furthermore, linear contracts possess good approximation properties, for instance, as studied in [3], which describes their results as follows:
> "First, we consider the case where the principal knows only the ﬁrst moment
> of each action’s reward distribution, and we prove that linear contracts are guaranteed to be worst-case
> optimal, ranging over all reward distributions consistent with the given moments. Second, we study
> linear contracts from a worst-case approximation perspective, and prove several tight parameterized
> approximation bounds."
>
> Furthermore, the work [4] describes how linear contracts are also robust. Gabriel Carroll describes her results as follows: "We consider a simple moral hazard problem, under risk-neutrality and limited
> liability, in which the principal is uncertain about the technology available to the
> agent. The principal knows some actions available to the agent, but other, unknown
> actions may also exist. The principal evaluates contracts according to their worst-
> case performance, with respect to the actions that may or may not be available
> to the agent. Under very general circumstances, the unique optimal contract is
> linear. This model thus provides a new explanation for the widespread use of linear
> contracts in practice, as well as a flexible and tractable modeling approach for moral
> hazard under non-quantifiable uncertainty."
>
> Furthermore, as mentioned by Gabriel Carroll, linear contracts are widely used in practice; for example, CEO compensation or sales commissions often have this structure. More heuristically, we would also argue that in a real-world setting, a linear contract giving a fixed percentage of the principal’s reward would be one of the most natural contracts to consider for both the principal and the agent.
> We will make it clearer in the next version of the paper that linear contracts are an important and natural form of contract.
>
> **Question 3:**
> > "In real-world applications ... It would be more interesting."
>
> and addressing the comment
>
> > "The methodology of empirical utility maximization ...1-dimensional parameter with
>  samples is not a challenging task."
>
> **Answer 3:**
>
>  We agree that the online setting is interesting and is studied, for instance, in [5], which obtains a regret bound of $ \tilde{O}(T^{2/3}) $, matching up to logarithmic factors a clean lower bound of $ \Omega(T^{2/3}) $ (page 3, Theorem 3 and the surrounding text in the arXiv version). Thus, they do not, as this work does, completely settle the statistical complexity of the problem, which to the best of our knowledge is also not the case for any other statistical setting studying linear contracts prior to our work. Even though we agree that the online setting is interesting, we think the batch setting is just as interesting and important since not all contracts are made in an online fashion, i.e., the agent does not always allow the principal to change the contract it offers repeatedly, or the other way around.
>
> [1] Multi-Agent Combinatorial Contracts: Paul Dütting et al.
>
> [2] The Pseudo-Dimension of Contracts: Paul Dütting et al.
>
> [3] Simple Versus Optimal Contracts:  Paul Dütting et al.
>
> [4] Robustness and Linear Contracts: Gabriel Carroll
>
> [5] The Sample Complexity of Online Contract Design: Banghua Zhu et al.

---

> > ### Author Rebuttal · Reviewer_DA5P · 2026-04-02
> >
> > Thanks for the explanation, but as exceedingly substantial parts of the paper need to be revised, including the comparison of bounds and justification of the significance of linear contracts. Hence, the paper is currently far from publication at this moment, so I choose to keep the original score.

---

> > > ### Author Response · Authors · 2026-04-03
> > >
> > > Dear Reviewer DA5P,
> > >
> > > We accept the reviewer’s view, but respectfully express that we disagree with the characterization that this constitutes a major revision or that the paper is far from publication.
> > >
> > > We already mention, albeit briefly, in the paper the appeal of the simplicity of linear contracts from a practical point of view (column 1, page 2), that linear contracts are one of the most fundamental classes of contracts (column 1, page 2), the approximation guarantees of [3] (column 1, page 2), the robustness guarantees of [4] (column 1, page 2), and the online setting [5] in the related work (page 3).
> > >
> > > Finally, the issue with the double citation identified in the first question is a simple matter of how the references are compiled and can be easily corrected.
> > >
> > > Thank you for engaging in the review process and for your feedback.

---

### Official Review · Reviewer_qHJF · 2026-03-12

**Soundness:** 3
**Presentation:** 3
**Significance:** 3
**Originality:** 3
**Overall Recommendation:** 5
**Confidence:** 4

**Summary:**

This paper studies the sample complexity of linear contract design.  The agent has a type $\theta$ that determines the agent's production functions (stochastic mapping from action to outcomes) and the cost associated with each action.  The principal observes samples of $\theta$ from an unknown distribution and aims to find an $\epsilon$ approximately optimal linear contract.  A linear contract shares $\alpha$ fraction of the reward with the agent, so the principal aims to find an $\epsilon$-optimal $\alpha^*$.  Previous work showed that this can be done with $O([\log(1/\epsilon) + \log(1/\delta)] / \epsilon^2)$  or $O([\ln(n) + \log(1/\delta)] / \epsilon^2)$ samples, where $n$ is the number of actions of the agent.  This paper claims to improve the sample complexity bound to $O(\log(1/\delta) / \epsilon^2)$, which should be tight if true.

**Compliance With Llm Reviewing Policy:**

Affirmed.

**Final Justification:**

See Rebuttal Acknowledgement.

**Key Questions For Authors:**

See the major soundness issue I mentioned above.  Did I misunderstand anything?

**Limitations:**

Minor typo:
* The $S$ in Algorithm 1 should be in bold font.

**Strengths And Weaknesses:**

_Major soundness issue_:

I don't think the theoretical result is correct.  Lemma 2.2 is problematic.  In the proof of Lemma 2.2 at the end of Page 7, the authors claim that $\sqrt{\sum_{i=1}^s | u_p(\theta_i, a) - u_p(\theta_i, \hat \alpha) |^2 / s} \le \sqrt{\sum_{i=1}^s | u_p(\theta_i, a) - u_p(\theta_i, \hat \alpha) | / s} \le \sqrt{\nu^2} \le \nu$.  The second inequality is not correct, because equation (10) only says $\sum_{i=1}^s | u_p(\theta_i, a) - u_p(\theta_i, \hat \alpha) | / s \le \nu$ but not $\le \nu^2$.  What we actually have is $\sqrt{\sum_{i=1}^s | u_p(\theta_i, a) - u_p(\theta_i, \hat \alpha) | / s} \le \sqrt{\nu}$, which is larger than the $\nu$ term claimed by the authors (because $\nu \le 1$).

I tried to derive the claimed inequality myself using the inequality $\sqrt{\sum_i |a_i|^2} \le \sum_i |a_i|$, but I only got $\sqrt{\sum_{i=1}^s | u_p(\theta_i, a) - u_p(\theta_i, \hat \alpha) |^2 / s} \le \sqrt{s} \nu$, which has an extra $\sqrt{s}$ term.  In fact, I think the extra $\sqrt{s}$ term is unavoidable.  Say we have a vector $x = (s\nu, 0, 0, ..., 0)$ with $s \nu \le 1$ (so it satisfies $|x_i| \le 1$), then $\sum_{i=1}^s |x_i| / s = \nu$ but $\sqrt{\sum_{i=1}^s |x_i|^2 / s} = \sqrt{s} \nu$, so the authors' claim of $\sqrt{\sum_{i=1}^s |x_i|^2 / s} \le \nu$ cannot be true.

This error is crucial because it may imply that the sample complexity is not $O( \log(1/\delta) / \epsilon^2)$ as claimed by the authors.

---

> ### Author Rebuttal · Authors · 2026-03-30
>
> Dear Reviewer qHJF,
>
> Thank you for reading and assessing our paper carefully. Furthermore, thanks for raising the concern about Lemma 2.2 and giving us the opportunity to explain why the proof is sound.
>
> **Concern 1:**
> >"See the major soundness issue I mentioned above. Did I misunderstand anything?"
>
> **Answer 1:**
> Our understanding is that the concern did not pertain to the following statement (Equation 10 and the text above it on page 7). We therefore assume that we have proven that (for clarity in the following, we have renamed $ \nu $ in Equation 10 and the text above to $ \nu' $ ):
>
>
> **Assumed Proven Start**
>
> For any $\nu' \in (0,1) $ there exists a set of contracts $ \mathcal{C}\_{\nu'} $
> which has size $ |\mathcal{C}\_{\nu'}|\leq 12/\nu' $
> and such that for any contract $ \alpha \in [0,1] $, there exists $ \hat{\alpha}\in \mathcal{C}\_{\nu'} $ which satisfies that
>
> \begin{align}
>     \textstyle\sum_{i=1}^{s}|u_{p}(\theta_i,\alpha)-u_{p}(\theta_i,\widehat{\alpha})|/s\leq \nu'.
> \end{align}
>
> **Assumed Proven End**
>
> We will now argue that having shown the above implies Lemma 2.2, which is (notice that the size is larger in the following lemma than in the statement of "Assumed Proven"):
>
> **Lemma 2.2**
>     For any $ 1>\nu>0 $ and $ S=(\theta\_{1},\ldots,\theta\_{s}) $, there exists a set of linear contracts $ \mathcal{C}\_{\nu}\subset [0,1] $ such that
>    - $ |\mathcal{C}\_{\nu}|=12/\nu^{2} $
>  - For any linear contract $ \alpha\in[0,1] $, there exists a contract $ \widehat{\alpha}\in \mathcal{C}\_{\nu} $ such that
>         \begin{align*}
>     \sqrt{\textstyle\sum_{i=1}^{s}|u_{p}(\theta_i,\alpha)-u_{p}(\theta_i,\widehat{\alpha})|^{2}/s}\leq\nu.
> \end{align*}
>
>
> Now to show Lemma 2.2, let $ \nu \in (0,1) $ be given. Since we showed the statement in, "Assumed Proven", for any $ \nu'\in (0,1) $, we can define $ \nu'=\nu^{2}$, which satisfies $\nu' \in(0,1) $, and invoke the statement in "Assumed Proven" with $ \nu' $ to obtain a set of contracts $ \mathcal{C}\_{\nu'} $ which has size $ |\mathcal{C}\_{\nu'}|\leq 12/\nu'=12/\nu^{2} $ and such that for any contract $ \alpha \in [0,1] $, there exists $ \hat{\alpha}\in \mathcal{C}\_{\nu'} $ which satisfies that
>
> \begin{align}
>     \textstyle\sum_{i=1}^{s}|u_{p}(\theta_i,\alpha)-u_{p}(\theta_i,\widehat{\alpha})|/s\leq \nu'=\nu^{2}. \quad (\star)
> \end{align}
>
> Thus we have that
>
> \begin{align*}
>     &\sqrt{\textstyle\sum_{i=1}^{s}|u_{p}(\theta_i,\alpha)-u_{p}(\theta_i,\widehat{\alpha})|^{2}/s}
>     \\
>     \underbrace{\leq}\_{by \\ |u_{p}(\theta_i,\alpha)-u_{p}(\theta_i,\widehat{\alpha})|\leq 1} &\sqrt{\textstyle\sum_{i=1}^{s}|u_{p}(\theta_i,\alpha)-u_{p}(\theta_{i},\widehat{\alpha})|/s}
>     \\
>     \underbrace{\leq}\_{by  \\ (\star)} &\sqrt{\nu^{2}}
>     \\
>     =&\nu,
> \end{align*}
>
> which is the claim of Lemma 2.2, where $ \mathcal{C}\_{\nu'} $ is renamed to $ \mathcal{C}\_{\nu} $.
>
> So, at a high level, since we invoke the statement in "Assumed Proven" with a higher precision, we also get a bound in $ L_{2} $-norm. The higher precision is accounted for in the larger size of the set, which is now $ 1/\nu^{2} $.
>
> Please let us know if the above clarifies the concern. Furthermore, again our understanding was that there were no concerns about the "Assumed Proven" part of Lemma 2.2. If there are concerns about the "Assumed Proven" part, we are happy to answer questions about this part of the proof also.
>
> If the concern is resolved by the above, and the concern had an impact on the initial assessment of the paper, we would be grateful if the review could be revisited.

---

> > ### Author Rebuttal · Reviewer_qHJF · 2026-04-03
> >
> > My misundersanding came from the misuse of notations.  The authors' response where they denote $\nu^2 = \nu'$ resolves my misunderstanding.
> >
> > I changed score from 2 to 5.
> >
> > This is a strong result.  A sample complexity bound like $O(\ln(1/\delta) / \epsilon^2)$ is rare in mechanism design problems.  It shows that the number of samples needed to learn the optimal linear contract (and even uniform convergence for all linear contracts) is at the same order as the sample complexity of learn the expected utility of a single linear contract.  This is a surprising result.
> >
> > I don't agree with the fourth reviewer's criticism that "The space of all linear contracts is only 1-dimensional. Learning an $\epsilon$-approximation of an 1-dimensional parameter with $\tilde O(1 / \epsilon^2)$ samples is not a challenging task."  The expected reward of a linear contract is not a Lipschitz continuous function of the 1-dimensional parameter.  Typical sample complexity bounds (proven by, e.g., pseudo-dimension) are $O( (d \ln(1/\epsilon) + \ln(1/\delta) ) /\epsilon^2)$ where $d$ is a complexity measure of the function.  This papers show that the linear contract function class, despite discontinuous, essentially has a constant complexity measure $d = O(1)$ and removes the $\ln(1/\epsilon)$ term in the bound. This is a significant contribution in my opinion.
> >
> > Finally, I suggest the authors add a conclusion/discussion section in the end to discuss some issues in the rebuttal.  For example, the authors' response to the first reviewer's Question 2 -- "Thus, our results can be seen as showing that if one can bound the $\nu$-cover directly instead of going through pseudo dimension, then one can get better and, in our case, tight bounds. We believe that taking this approach for other simple classes of contracts could also lead to tight bounds."-- can be included and highlighted.

---

> > > ### Author Response · Authors · 2026-04-03
> > >
> > > Again thank you for your active participation in the review process.
> > >
> > > We agree that we should incorporate as much feedback and discussion from the rebuttal as possible into the next version of the paper, much of which would fit naturally in a concluding discussion section.

---

### Official Review · Reviewer_xkkc · 2026-03-12

**Soundness:** 3
**Presentation:** 3
**Significance:** 3
**Originality:** 3
**Overall Recommendation:** 5
**Confidence:** 4

**Summary:**

The paper provides the sample complexity for a setting where a principal is trying to learn its utility function (or equivalently the agent population's type distribution) when interacting one-on-one with a population of agents. These interactions are one-shot interactions in a Bayesian game setting, where principal sets a linear contract (payment function), and the agent observes the payment function and takes an action, and a random outcome arises from the contract-action pair.

**Compliance With Llm Reviewing Policy:**

Affirmed.

**Key Questions For Authors:**

None

**Limitations:**

Yes

**Strengths And Weaknesses:**

This paper is can be made sound by massaging some of the assumptions: why does the agent break ties in favor of the principal? Why do we care about sample complexity (number of samples of the type) when the principal only has access to the oracle? Should we be talking about oracle-access complexity?

Presentation is well structured, but there are gaps in the writing: you didn't define the U_p(S,alpha) function in the main result, some of the important parts of the model are not clearly stated, e.g., "known reward", is it known to every players? How can this be known to the agent in an example like Spotify? Any connection between EUM and empirical risk minimization?

It is a significant problem with application Spotify, it also highlights a connection to classical sample complexity results in VC theory, although there is no explicit mention of this. It could definitely influence a bunch of follow-up works in this general direction.

Originality: the paper cites a line of works on algorithmic contract theory, but the application of sample complexity analyses in that line seems new and noteworthy.

---

> ### Author Rebuttal · Authors · 2026-03-30
>
> Dear Reviewer xkkc,
>
> Thank you for reading and assessing our paper carefully. We will now address the questions to the paper.
>
> **Question 1:**
> >"Why does the agent break ties in favor of the principal?"
>
> **Answer 1**
>
> The reason for our choice of this tie-breaking rule is that, to the best of our knowledge, it is the standard in the literature; see e.g., [1] (Top of page 7), [2] (Footnote page 4 with a comment about it being standard), [3] (middle page 3), [4] (top page 9), [5] (top page 7).
> The latter ([5]) has the following comment in the footnote explaining the reasoning for this choice:  "The idea is that one could perturb the payment schedule slightly to make the desired action uniquely optimal for the agent. For further discussion see [11, p. 8].". We were unfortunately not able to access the reference [11, p. 8] in [5]. We will add a comment about this choice being standard in the next version of the paper.
>
>
> **Question 2:**
> >"Why do we care about sample complexity (number of samples of the type) when the principal only has access to the oracle? Should we be talking about oracle-access complexity?"
>
> **Answer 2:**
>
> We think oracle-access complexity is an interesting question and worth studying in its own right. In this paper, we wanted to study the information-theoretic complexity of the problem, namely the sample complexity: how many samples are needed for what we see empirically to generalize to the true underlying distribution from which the empirical observations came. Assuming the oracle allows us to detach these two problems and focus on the information-theoretic one.
>
> **Question 3:**
>
> >"Presentation is well structured, but there are gaps in the writing: you didn't define the $U_p(S,\alpha)$ function in the main result, some of the important parts of the model are not clearly stated, e.g., "known reward", is it known to every players? How can this be known to the agent in an example like Spotify?"
>
> **Answer 3:**
>
> We agree that we should have defined $ U_{p}(S,\alpha) $ properly and made it clear that the reward is known to both the principal and the agent, following the work of [1]. We thank the reviewer for pointing this out. We will change this in the next version of the paper and read carefully through the paper to clean up the writing further.
> We agree that in the Spotify example, the agent would not know the reward. However, from a modeling perspective, this is still captured by our work, since an oblivious agent, i.e., an agent not acting on the reward function, is captured within the larger set of possible agents who may act based on the reward, implying that our uniform convergence bounds also hold here for each fixed reward vector $ r $.
>
> **Question 4:**
> >"Any connection between EUM and empirical risk minimization?"
>
> **Answer 4:**
> That is a great point - yes, there are connections to Empirical Risk Minimization and the guarantees obtained for it in Learning Theory, as it is the negation of the Empirical Risk Maximization problem we consider in the paper. We will add a section about this in the next version of the paper, making this clearer.
>
>
> [1]: The Pseudo-Dimension of Contracts: Paul Dütting, Michal Feldman, Tomasz Ponitka, Ermis Soumalias (arxiv version)
>
> [2]: Bayesian Agency: Linear versus Tractable Contracts: Matteo Castiglioni, Alberto Marchesi, Nicola Gatti (arxiv version)
>
> [3]: Optimal Common Contract with Heterogeneous Agents: Shenke Xiao, Zihe Wang, Mengjing Chen, Pingzhong Tang, and Xiwang Yang (arxiv version)
>
> [4] Algorithmic Contract Theory: A Survey: Paul Dütting, Michal Feldman, Inbal Talgam-Cohen
>
> [5] Simple versus Optimal Contracts: Paul Dütting, Tim Roughgarden, Inbal Talgam-Cohen

---

> > ### Author Rebuttal · Reviewer_xkkc · 2026-04-01
> >
> > My concerns were resolved, but the third reviewer pointed out something perhaps even more important.

---

### Official Review · Reviewer_HNZX · 2026-03-14

**Soundness:** 4
**Presentation:** 4
**Significance:** 3
**Originality:** 4
**Overall Recommendation:** 5
**Confidence:** 3

**Summary:**

This paper explores the problem of learning optimal linear contracts when the principal has oracle access to computing the empirical utility of a candidate contract. The authors show that the simple algorithm of choosing the contract that maximizes the principal's empirical utility achieves an epsilon-approximation to the optimal contract with high probability, assuming sufficient samples are available. The authors provide a tight characterization of the sample requirements, effectively settling this problem for linear contracts. The authors also provide a uniform convergence guarantee that establishes that the empirical utility of every linear contract is an epsilon approximation of its true expectation. These results improve upon the sample complexity results of (Dütting et al. 2015) for linear contracts by exploiting the unique structure of linear contracts.

**Compliance With Llm Reviewing Policy:**

Affirmed.

**Key Questions For Authors:**

1. This was briefly touched upon in the paper, but could you expand on how much weaker your oracle access assumption is compared to the original assumption in (Dütting et al. 2015)? Do you know what results from (Dütting et al. 2015) hold under this weaker assumption?

2. How specific are are your proof techniques to the class of linear contracts? Do you expect any of these tools to be useful for proving sample complexity bounds for other simple classes?

**Limitations:**

It is unclear to me, whether the proof techniques can be used to help provide sample complexity bounds for other classes of contracts. In contrast, the notion of the Pseudo-Dimension of contracts (Dütting et al. 2015) appears to provide a very general framework for proving sample-complexity results for contracts.

**Strengths And Weaknesses:**

- The paper is appears technically sound. The proofs are well-explained and rely on milder assumptions than existing works.

- The paper is well presented overall, and sufficient explanation is provided as to its relation with (Dütting et al. 2015), which established the previous (loose) bound on sample complexity. A few sections should be rewritten for greater clarity. For example, I do not follow the sentence in lines 127-135 beginning with “To the best of our knowledge….”.

---

> ### Author Rebuttal · Authors · 2026-03-30
>
> Dear Reviewer HNZX,
>
> Thank you for reading and assessing our paper carefully. We will now address each of the two questions from the reviewer:
>
> **Questions 1:**
> >"This was briefly touched upon in the paper, but could you expand on how much weaker your oracle access assumption is compared to the original assumption in (Dütting et al. 2015)? Do you know what results from (Dütting et al. 2015) hold under this weaker assumption?"
>
> **Answer 1**
>
> The results in Dütting et al. 2025 for the linear contract case also use the EUM algorithm, which only needs this weaker oracle access. For their other results, they use the definition of an Approximation Oracle (Definition 5.1 in [1]), which takes as input the full knowledge of the agent. They write "To derive sample- and time-efficient algorithms, we introduce the concept of an approximation oracle,
> which provides a near-optimal contract for an empirical distribution of agent types. This approach
> decouples the learning problem from the computational challenge of identifying the optimal contract
> for a given (known) distribution. For many contract classes and distributions, such oracles can be
> constructed [e.g., 13, 38, 1], and we will demonstrate how to leverage them effectively". Thus they take a hands-off approach to the computational aspect of the problem (which the references [13,38,1] in Dütting et al. 2025 consider, and which is an important line of research); in order to focus on the learning problem, so do we.
>
>
> **Questions 2:**
> >"How specific are are your proof techniques to the class of linear contracts? Do you expect any of these tools to be useful for proving sample complexity bounds for other simple classes?"
>
> **Answer 2:**
>
> The chaining framework we use in the paper works as soon as one has a bound on a $ \nu $-cover, so the crux of getting good bounds for new classes (and possibly optimal ones) is designing such small covers. For some context on what is done in Dütting et al. 2025, they bound the pseudo dimension of different contract classes, and since a bound on the pseudo dimension gives a bound on the $ \nu $-cover, their results follow by the chaining framework. Thus, our results can be seen as showing that if one can bound the $ \nu $-cover directly instead of going through pseudo dimension, then one can get better and, in our case, tight bounds. We believe that taking this approach for other simple classes of contracts could also lead to tight bounds.

---

> > ### Author Rebuttal · Reviewer_HNZX · 2026-04-03
> >
> > Thank you for your response, both the answers make sense to me.

---

### Decision · Program_Chairs · 2026-04-30

**Decision:**

Accept (regular)

**Comment:**

The paper studies the sample complexity of learning an optimal linear contract in Bayesian principal-agent problems in which the agent's type is drawn at each round from an unknown fixed probability distribution. The paper gives a tight characterization of the sample requirements, and it also provides a uniform convergence guarantee. These results improve upon the sample complexity results of (Dütting et al. 2025) by exploiting the unique structure of linear contracts.

The Reviewers are generally positive on this paper, with the only exception of Reviewer DA5P, which is slightly negative, arguing that the novelty and significance of this paper are not strong enough or not well justified. However, all the Reviewers agree that the paper is technically correct and well written. I personally believe that the paper closes some gaps in the literature on learning optimal contracts in principal-agent problems, and it would be a nice fit for the conference. Thus, I suggest acceptance of the paper.